# Phytochemical Profile and In Vitro Bioactivities of Plant-Based By-Products in View of a Potential Reuse and Valorization

**DOI:** 10.3390/plants12040795

**Published:** 2023-02-10

**Authors:** Ilaria Chiocchio, Manuela Mandrone, Massimo Tacchini, Alessandra Guerrini, Ferruccio Poli

**Affiliations:** 1Department of Pharmacy and Biotechnology, Alma Mater Studiorum–University of Bologna, Via Irnerio 42, 40126 Bologna, Italy; 2Department of Life Sciences and Biotechnology, University of Ferrara, Via Borsari 46, 44100 Ferrara, Italy

**Keywords:** plant by-products valorization, circular economy, tyrosinase inhibition, antioxidant, phytopathogens

## Abstract

Wastes and by-products of plant origin are of particular interest to develop a circular economy approach, which attempts to turn them into resources. In this work, thirty-seven neglected plant matrices, including agricultural residues, pest plants, and by-products from the herbal and food industry were extracted and tested for their in vitro anti-tyrosinase, antioxidant, and antibacterial activity against the phytopathogens *Pseudomonas syringae* pv. *syringae* ATCC 19310 and *Clavibacter michiganensis* subsp. *nebraskense* ATCC 27822. Antioxidant activity ranged from 0.3 to 5 mg of Tr. eq/mL of plant extract, and extract of *Castanea sativa* pericarp (Csp), *Rosa damascena* buds (post-distillation) (Rod), and *Prunus amygdalus* exocarp and mesocarp (Pam) were the most powerful ones. Csp was also capable of inhibiting tyrosinase (IC_50_ = 16.5 µg/mL), as well as three distillation by-products, namely: *Cupressus sempervirens* (Css) (IC_50_ = 95.5 µg/mL), *Salvia officinalis* (Sco) (IC_50_ = 87.6 µg/mL), and *Helichrysum italicum* (Hei) (IC_50_ = 90.1 µg/mL). Five residues from distillation showed antibacterial activity against *C. michiganensis* (MICs ranging from 0.125 to 1 mg/mL), namely: *Salvia sclarea* L. (Sas), *Salvia rosmarinus* Schleid (Sar), Sco, Hei, and Css. The ^1^H NMR fingerprinting of the bioactive matrices was acquired, detecting primary and secondary metabolites (rosmarinic acid, shikimic acid, sclareol, and hydroxycinnamic acids).

## 1. Introduction

The old economic approach, based on the logic of “make, use, dispose”, is no longer sustainable and needs to be replaced. One of the current alternatives is represented by the circular economy, which aims to create closed loops in industrial systems by both minimizing waste/by-products and turning them into resources [1]. The transition to this kind of economy system requires actions and policy, and coherently the European Commission adopted the first circular economy action plan in 2015 and a new one in 2019 [2], which lists key products such as electronics, plastics, and food wastes.

In order to implement this strategy, in the past two decades, the scientific community has been paying increasing attention to the valorization of industrial and agricultural by-products of plant origin [3,4]. These biological matrices are particularly interesting since they are a source of high-value compounds such as secondary metabolites. These compounds are, in fact, endowed with numerous biological activities, which make them useful in different fields such as cosmetic, nutraceutical, pharmaceutical, food additives, and antifeedant [5]. In this context, thirty-seven neglected plant matrices were tested as in vitro tyrosinase inhibitors, antioxidants, and antibacterial against phytopathogens.

Tyrosinase inhibitors are candidates for skin-whitening agents that are useful in cosmetics to counteract age-related skin hyperpigmentation, and they also have potential in the treatment of diseases associated with tyrosinase misregulated expression and/or activity, such as urticaria pigmentosa [6]. In fact, tyrosinase is responsible for the rate-limiting first two steps of the melanin biosynthetic pathway, and thus, for skin, hair, and eye color in humans [7]. Moreover, tyrosinase inhibitors also find an application as preservatives, since they are capable of slowing down the food browning process mediated by the activity of this enzyme [8]. Hence, natural tyrosinase inhibitors represent an alternative approach for medicinal and cosmetic products, as well as in the food industry. According to a recent review [9], the majority of the scientific studies published between 2006 and 2020 deal with the potential use of neglected plant matrices as a source of bioactive molecules for human health care, while there are only few reports about the use of these sources in agriculture.

Agriculture itself is one of the most challenging sectors for sustainable development. Indeed, population growth and the consequent rising demand for food require increasingly higher yields. However, lands are limited, and overharvesting has caused several environmental problems such as depletion and pollution of water, hydrologic modifications, emission of greenhouse gasses, and degradation of soil quality and fertility due to the use of fertilizers and pesticides [10,11]. Hence, more sustainable farming techniques need to be implemented and adopted. In this context, plant-derived compounds might represent a valuable alternative to the use of synthetic compounds. In this work, the samples were tested against strains of *Pseudomonas syringae* pv. *syringae* ATCC 19,310 and *Clavibacter michiganensis* subsp. *nebraskense* ATCC 27822. *Pseudomonas syringae* pv. *syringae* van Hall (ATCC 19310) was chosen as Gram-negative bacteria because it is the most polyphagous in the *P. syringae* complex that primarily affects woody and herbaceous host plants. Alongside this microorganism, the bacterium *Clavibacter michiganensis* subsp. *nebraskense* (ATCC 27822) was chosen, as it is a Gram-positive bacterium that affects the maize plant (the third most cultivated cereal in the world with an increasing cultivation trend [12]) during all its growth stages.

Hence, the present work aims at valorizing neglected plant matrices, including agricultural residues, pest plants, and herbal and food industry by-products, investigating the in vitro antioxidant and anti-tyrosinase activity together with the antibacterial activity against common phytopathogens. In addition, the total phenolic and flavonoid contents of the samples were also determined. Finally, preliminary information about the phytochemical composition of the most active samples was obtained through ^1^H NMR profiling.

## 2. Results and Discussion

As reported in Table 1, the tested plant matrices were classified according to their origin as agricultural residues (15 matrices), pest plants (5 matrices), solid waste from distillation (12 matrices), and food industry by-products (5 matrices).

The fifteen agricultural by-products include plant organs, which are discarded because they are inedible, and this is generally the case of leaves or aerial parts of many crops, such as onion, sugar beet, bean, chickpea, courgette, camelina, sunflower, tomato, potato, wheat, and sorghum. These plant parts are generally understudied compared to edible organs. However, some of these matrices have already shown an interesting pattern of bioactivities; for instance, onion leaf extract was found to ameliorate diabetes-induced neuropathic pain [13]. Tomato and potato leaves contain several glycoalkaloids with a wide spectrum of biological activities, among which anti-inflammatory, antinociceptive, antipyretic, and antiproliferative effects [14]. Beet root leaves contain several bioactive secondary metabolites such as flavonoids, saponins, betalains, and carotenoids showing several biological activities, including antioxidant, hepato-protective, and chemo-preventive [15]. Leaves of cereals, such as sorghum and wheat, also contain health-promoting compounds. The most prominent among them are phenolics and flavonoids [16] and, in particular, wheat leaves that contain the flavonoid tricin, which is endowed with numerous health-beneficial effects such as antioxidant, immunomodulatory, antiviral, anti-ulcerogenic, and potential antitumor/anticancer activity [17].

*Castanea sativa* spiny burs were also included in this study since they are an important by-product of chestnut cultivation. However, they were found to be endowed with neuroprotective potential, antioxidant and antibacterial activity, and rich in tannins and other bioactive compounds [18,19].

In addition to neglected crop organs, pest plants contribute to the generation of another kind of important agriculture by-product. In this work, different organs from common pest plants, namely: *Abutilon theophrasti*, *Cichorium intybus*, *Echinochloa crus-galli*, and *Erigeron canadensis*, were analyzed. Despite being regarded as weeds, many of these plants are also medicinal plants, used, for instance, in traditional medicine as they are rich in bioactive compounds. For example, *Abutilon theophrasti* is used in Chinese traditional medicine as an expectorant, diuretic, analgesic, anti-inflammatory, anthelmintic, demulcent, aphrodisiac, and emollient [20]. *Cichorium intybus* is reported to contain gallic acid, protocatechuic acid, chicoric acid, chlorogenic acid, caftaric acid, flavonoids, inulin, sesquiterpene lactones, coumarins, and/or other compounds, conferring numerous bioactivities [21] to this plant.

The twelve by-products from the herbal industry investigated in this work were all solid residues from the distillation of leaves or aerial parts of common aromatic plants, namely: laurel, lavender, lemon balm, thyme, sage, clary sage, rosemary, yarrow, wormwood, cupressus, curry plant, and damask rose. Every year, the essential oil industry generates large amounts of waste due to the low content of volatiles in the fresh plant matrices (less than 5% (*w*/*w*) of the plant material used). Most of the distilleries simply discard the waste biomass even though it might be a valuable source of bioactive polyphenols useful for the pharmaceutical, cosmetic, and food industries [22,23].

Residues from the food industry include different plant parts discarded during the production of food and beverages, such as grape pomace, almond exocarp and mesocarp, chestnut pericarp, bean husk, and oregano stems. Some of these by-products are rich in polyphenols, offering an opportunity for their valorization [24,25].

Although the plant matrices studied in this work are very diverse, the same protocol of extraction, based on the hydroalcoholic solvent system, was chosen. In fact, previous studies [26,27] report that a mixture of methanol and water represents one of the best extraction conditions to obtain a broad spectrum of compounds from generic plant samples.

The antioxidant activity of the samples measured by means of DPPH assay ranged from 0.3 to 5 mg of Tr. eq/mL of plant extract. Extract of *Castanea sativa* pericarp (Csp), *Prunus amygdalus* exocarp and mesocarp (Pam), and *Rosa damascena* buds (post-distillation) (Rod) were the most powerful ones (Figure 1 and Appendix A), exerting an antioxidant activity equal to 4.76 ± 0.16, 5.1 ± 0.07, and 4.77 ± 0.12 mg of Tr. eq/mL, respectively, all significantly higher than the other samples.

All plant extracts were analyzed for their total phenolic and flavonoid content (Figure 2; Appendix A). Total phenolic content ranged from 2 to 44 mg of GAE/g of dried plant material (DW) while total flavonoid content ranged from 0 to 43 mg of RE/g of DW. Except for few samples having very low phenolic and flavonoid content (below 5 mg GAE/g and 5 mg RE/g), namely aerial parts of *Camelina sativa* (L.) (Csa), *Allium cepa* (Ace), and apical flowering aerial parts of *Cichorium intybus* L. (Cia), most of the examined matrices were still a valuable source of these antioxidant compounds.

Csp together with Rod and Pam were the matrices with the highest antioxidant activity, and they were also the richest in phenolic compounds (concentration higher than 20 mg GAE/g (DW)), consistently with the renowned antioxidant capacity associated to these phytochemicals. Rod was also one of the matrices with the highest content of flavonoids (35.34 mg RE/g (DW)), while Csp and Pam were among the samples with a low flavonoid content (4.8 and 3.7 mg RE/g (DW)) compared to the others.

Previous studies explored the possibility of valorizing chestnut pericarp through the investigation of its polyphenolic profiles. Low molecular weight phenolics (gallic and ellagic acid), condensed tannins, and ellagitannins were found in the pericarp and integument of Portuguese cultivars of *Castanea sativa* [28]. In addition, Pinto et al. [25] found that metabolites from chestnut pericarp are also released in wastewater after water curing, a traditional practice to prevent insect and mould development during fruit storage. *Prunus amygdalus* mesocarp, referred to as hulls, is a natural source for sweetener concentrate and dietary fiber, and it contains triterpenoids, lactones, and phenolics [29,30,31].

A recent study on *Rosa damascena* [32] proved that methanolic extracts of petals collected at different development stages contain phenolic compounds such as catechin, epicatechin, gallic acid, chlorogenic acid, and naringin, which are all known antioxidant compounds. These compounds are not extracted by distillation, which targets only the essential oil, leaving them to the waste biomass. In fact, in our study, Rod was endowed with in vitro antioxidant activity, and it was still rich in both phenols and flavonoids.

The antioxidant activity values of Hei (*Helichrysum italicum*), Meo (*Melissa officinalis*), Css (*Cupressus sempervirens*), and Sar (*Salvia rosmarinus*) were statistically comparable, and these matrices were the most potent after Pam, Rod, and Csp. Interestingly, Hei, Meo, Css, and Sar are all residues after distillation, confirming once more the richness in bioactive compounds of these by-products.

All extracts, at a concentration of 100 µg/mL, were tested as tyrosinase inhibitors, and the ones giving a percentage of inhibition higher or equal to 30% were considered promising, thus their IC_50_ was further calculated. All the percentages of tyrosinase inhibitory activity found at 100 µg/mL are reported in Appendix A. Four extracts exhibited IC_50_ lower than 100 µg/mL, namely: Css (IC_50_ = 95.5 µg/mL), Hei (IC_50_ = 90.1 µg/mL), Sco (IC_50_ = 87.6 µg/mL), Csp (IC_50_ = 16.5 µg/mL).

*Castanea sativa* by-products were recently investigated for their possible valorization in cosmetics, and it was found that pericarp subjected to autohydrolysis exerted the best anti-tyrosinase activity (IC_50_ = 387 μg/mL) in comparison to burs and leaves [33]. In our study, we found the strongest activity for Csp, supporting that the hydroalcoholic solvent favors the extraction of the bioactive molecules. Different species of *Helichrysum* are known to be active against tyrosinase [34], and according to Gonçalves et al. [35], a methanolic extract of *H. italicum* leaves at a concentration of 10 mg/mL showed 74.13% of tyrosinase inhibitory activity. In this work, we have much refined this data by calculating the IC_50_ of Hei; moreover, we established that *Helichrysum italicum* after distillation is still active as a tyrosinase inhibitor. *Cupressus sempervirens* was also found active against tyrosinase by Zenigin et al. [36], who chose to express the activity in a less common unit of measure, namely mg of kojic acid equivalents/g of extract (KAE/g), and found an IC_50_ of 29.43 KAE. Using the same unit of measure, Css had an activity of 33.41 KAE. The tyrosinase inhibitory activity of *Salvia officinalis* leaves extract was already known, showing IC_50_ of 20.43  ±  1.39 µg/mL [37]. Thus, our work proves that after distillation, these by-products still retain good potential as tyrosinase inhibitors.

As shown in Appendix A, none of the tested extracts were active against the Gram-negative bacterium (*P. syringae* pv *syringae*), while thymol, used as a positive control, exhibited a MIC value of 0.0625 mg/mL. Regarding the results on the Gram-positive bacterium (*C. michiganensis* subsp. *nebraskense*), extracts from five plant wastes showed antibacterial activities with MICs ranging from 0.125 to 1 mg/mL. Interestingly, all the active matrices were solid residues from distillation, namely aerial parts of *Salvia sclarea* L. (Sas), *Salvia rosmarinus* Schleid (Sar), *Salvia officinalis* L. (Sco), and *Helichrysum italicum* (Roth) G. Don (Hei) and leaves of *Cupressus sempervirens* L. (Css). The extracts deriving from plants belonging to the genus *Salvia* showed antibacterial activity with MICs ranging between 0.5 and 1 mg/mL (Appendix A). Hei was the most active extract, with a MIC of 0.125 mg/mL, while positive controls thymol and Heliocuivre S. had a MIC of 62.5 µg/mL and 0.65625 µL/mL, respectively.

Gram-negative bacteria were less sensitive than Gram-positive bacteria, possibly because of the different bacterial wall characteristics that distinguish bacteria in Gram staining.

Samples showing significant antibacterial and anti-tyrosinase activity were analyzed by ^1^H NMR profiling in order to obtain an overview of their main phytochemicals (Figure 3). The analysis showed, in all plant extracts belonging to the genus *Salvia*, the presence of rosmarinic acid, whose antibacterial activity is well established [38]. Interestingly, the amount of rosmarinic acid in *S. sclarea* extract (Sas) was lower than the other two species (Appendix A), while this sample was found particularly rich in sclareol (57.39 µg/mg of dried plant matrix), a diterpene generally found in the essential oil of this plant. The amount of rosmarinic acid in the samples from diverse species of *Salvia* also reflected their antioxidant activity and total phenolic content. In fact, while Sco (*Salvia officinalis*) and Sar (*Salvia rosmarinus*) showed no significant differences in antioxidant activity and phenolic content, Sas (*Salvia sclarea*) was one of the less antioxidant samples, and its phenolic content was significantly lower than the other two species. The flavonoid content of Sas was also lower than Sar and Sco.

The activity of rosmarinic acid as a tyrosinase inhibitor is also reported [39]; however, the intensity of the spectral signals related to this compound in Sco and Sar ^1^H NMR profile is very similar, but only Sco is able to inhibit tyrosinase; thus, the overall inhibitory activity is not easily ascribable to rosmarinic acid. Residue from the distillation of the *C. sempervirens* (Css) ^1^H NMR profile showed the presence of shikimic acid along with other unknown aromatic and aliphatic compounds (at low concentration). The most powerful antibacterial sample was the extract of *Helichrysum italicum* (Hei), which was rich in hydroxycinnamic acids.

## 3. Materials and Methods

### 3.1. Chemicals

All reagents were purchased from Sigma Aldrich (Milano, Italy), except the deuterated solvents, which were purchased from Eurisotop (Cambridge, UK).

### 3.2. Plant Material and Sample Treatment

The matrices (Table 1) were provided by local producers (Emilia-Romagna region, Italy) including farmers, herbalists, and winemakers, and vouchers of the dried plant material were deposited in the Department of Pharmacy and Biotechnology, University of Bologna (via Irnerio 42, Bologna, Italy). Solid wastes from distillation, grape pomace, and almond exocarp and mesocarp were freeze-dried, while the other plant matrices were dried at 40 °C on the stove. After the drying process, all samples were grounded and stored in the dark and at room temperature.

### 3.3. Extracts Preparation

Extracts were prepared as described by Cappadone et al. [40] with slight modifications. Briefly, 120 mg of dried and powdered plant material underwent ultrasonic-assisted extraction for 30 min using 6 mL of MeOH/H_2_O (1:1). Subsequently, samples were centrifuged (2469× *g*) for 20 min, and each supernatant was separated from the pellet and divided into four tubes. The solvent was evaporated in a vacuum concentrator (Savant SpeedVac SPD210, Thermo Fisher Scientific, Waltham, MA, USA) in order to yield the crude extracts. Aliquots of the extracts were in turn used for ^1^H NMR profiling, anti-tyrosinase, and antimicrobial activity.

### 3.4. Total Phenolic Content and Total Flavonoid Content

Total polyphenol and flavonoid content and antioxidant activity were measured on the extracts before evaporating the solvent; the supernatant was separated from the pellet and directly tested.

The total phenolic and total flavonoid content of the extracts were assessed by means of spectrophotometric methods using the spectrophotometer Victor™ X3 PerkinElmer (Waltham, MA, USA) as described by Chiocchio et al. [41] with slight modifications. The extracts were diluted in methanol (1:1) and tested in duplicate for each assay. Gallic acid and rutin were used to build the calibration curves which were used to calculate by interpolation the total phenolic content and total flavonoid content, respectively. Thus, the total phenol content was expressed as mg of gallic acid equivalent (GAE) per g of dried plant material, and the total flavonoid content was expressed as mg rutin equivalent (RE) per g of dried plant material. The tests were repeated three times.

### 3.5. Total Antioxidant Activity

In vitro antioxidant activity of the extracts was determined using the DPPH free radical scavenging assay as described by Marrelli et al. [42] with slight modifications. Extracts were diluted in methanol in order to test different concentrations (ranging from 1.25 to 1000 μL/mL in the assay). Fifty μL of sample solutions were added to 700 μL of a DPPH methanol solution (50 mM). After 20 min at room temperature, absorbances (Abs) were measured at 517 nm and the percentage of antioxidant activity was calculated using the following formula:Antiradical activity % = ((control Abs − sample Abs)/control Abs) × 100(1)

Methanol was used as a negative control, Trolox (Tr) at different concentrations (ranging from 50 to 500 μM) was used as a positive control, and Tr IC_50_ was used for the calculation of Trolox equivalents. Total antioxidant activity was expressed as mg of Trolox equivalent (Tr. eq) per mL of extract. The tests were repeated three times.

### 3.6. Tyrosinase Inhibitory Assay

The enzymatic inhibitory assay was performed according to Chiocchio et al. [40] with slight modifications. Mushroom tyrosinase (2 mU) and a sample (50 μg/mL) were incubated for 5 min in 0.1 M sodium phosphate buffer pH 6.8, in 0.1 mL of final volume. L-DOPA (final concentration 2 mM) was added up to a final reaction volume of 0.2 mL. The formation of dopachrome was immediately monitored for 5 min at 490 nm in a microplate reader (Victor^™^
*X*3 PerkinElmer, Waltham, MA, USA) under a constant temperature of 30 °C. The IC_50_ (concentration necessary for 50% inhibition of enzyme activity) was calculated by constructing a linear regression curve showing extract concentrations (from 10 to 250 μg/mL) on the *x*-axis and percentage inhibition on the *y*-axis. A negative control was obtained by adding water instead of extracts, while kojic acid (solubilized in water) was used as a positive control, finding an IC_50_ of 3 ± 0.37 μg/mL (21 μM). The percentage of enzyme inhibition was calculated using the following formula:%Inhibition = [1 − (ΔAbs/min_sample_/ΔAbs/min_negative control_) × 100](2)

In order to determine the kinetic parameters for the enzymatic reaction, the Lineweaver-Burk plot was built, using a substrate concentration in the range from 0.5 to 4 mM. In the assay conditions, the obtained K_M_ value was of 0.2 mM and V_max_ of 10 μmol/min (ΔAbs/min = 0.03), considering dopachrome ε at 490 nm = 3.6201 mM^−1^ cm^−1^ and a light path length of 0.8 cm. The tests were repeated three times.

### 3.7. Antibacterial Activity

The antibacterial activity was evaluated against phytopathogenic bacteria to preliminarily verify phytoiatric and health properties. *Pseudomonas syringae* pv. *syringae* ATCC 19,310 and *Clavibacter michiganensis* subsp. *nebraskense* ATCC 27,822 were used to determine MIC (Minimum Inhibitory Concentration) through the microdilution method using 96-well microtiter plates.

Bacterial cultures were incubated overnight at 26 °C and 28 °C, respectively, in Tryptic Soy Broth (OXOID Ltd., Hampshire, UK). One hundred μL of sterile medium were used together with 100 μL of sample to perform serial dilutions of extracts previously dissolved in fresh medium (50 mg/mL of stock solution), into all micro-wells.

One hundred microliters of bacterial culture standardized to 2 × 10^7^ CFU/mL were added to the wells and incubated at 26 °C and 28 °C for 24 h. After the incubation period, 40 μL of water solution (20 mg/mL) of 2,3,5-triphenyl-tetrazolium chloride (Sigma-Aldrich, St. Louis, MO, USA) were added to each well and then incubated for 30 min: microbial growth was evaluated by a microplate reader (680XR, Bio-Rad, Laboratories, Inc., Hercules, CA, USA) at 415 nm. Thymol (concentration range 0.0625–0.5 μg/mL) and Heliocuivre S (commercial product; concentration range 0.65625–5.25 μL/mL) were used as a positive control. All determinations were made in triplicate.

### 3.8. NMR Analysis

For ^1^H NMR profiling, each extract was solubilized in deuterated solvents obtaining a final concentration equal to 5 mg/mL. The solvent used was a mixture (1:1) of phosphate buffer (90 mM; pH 6.0) in H_2_O-*d*_2_ containing 0.01% trimethylsilylpropionic-2,2,3,3-*d*_4_ acid sodium salt (TMSP) and MeOH-*d*_4_.

^1^H NMR spectra were recorded at 25 °C on a Varian Inova 600 MHz NMR instrument (600 MHz operating at the ^1^H frequency) equipped with an indirect triple resonance probe. Methanol-*d4* was used for internal lock. Each ^1^H-NMR spectrum consisted of 256 scans (corresponding to 16 min) with the relaxation delay (RD) of 2 s, acquisition time of 0.707 s, and spectral width of 9595.8 Hz (corresponding to δ 16.0). A presaturation sequence (PRESAT) was used to suppress the residual water signal at δ 4.83 (power = −6 dB, presaturation delay 2 s). The spectra were manually phased and the baseline corrected and calibrated to the internal standard trimethyl silyl propionic acid sodium salt (TMSP) at δ 0.0 using Mestrenova 14.3.1 software (Mestrelab Research, Spain). Compound identification was based on an in-house library and compared with data from the literature [43]. Semi-quantitative analysis of metabolites in the crude extracts was calculated by integration and comparison of diagnostic signals of the compounds (Appendix A) and TMSP (internal standard) resonating at δ 0.

### 3.9. Statistical Analysis

Values were expressed as the mean ± SD of one experiment performed in duplicate and repeated three times. Statistical analyses were performed using Graph Pad Prism 4 software (La Jolla, CA, USA). Samples were compared by one-way analysis of variance (ANOVA), followed by Tukey’s honestly significant difference (HSD) post-hoc test, considering significant differences at *p* values < 0.05.

## 4. Conclusions

In this screening of plant-neglected matrices, five out of thirty-seven samples were found active against strains of Gram-positive bacterium *C. michiganensis* subsp. *Nebraskense*, namely: *Salvia sclarea* L. (Sas), *Salvia rosmarinus* Schleid (Sar), *Salvia officinalis* L. (Sco), and *Helichrysum italicum* (Roth) G. Don (Hei) and leaves of *Cupressus sempervirens* L. (Css). Css, Hei, and Sco were found active also as tyrosinase inhibitors, together with *Castanea sativa* Mill pericarp (Csp). Most of the active samples were solid wastes after the distillation process, highlighting the potential of this kind of by-product to be re-used. In addition, twenty-three samples proved to exert in vitro antioxidant activity by means of DPPH assay, and the most powerful antioxidants were generally rich in phenolics and flavonoids. The ^1^H NMR profile revealed that these matrices are still rich in metabolites such as rosmarinic acid, shikimic acid, sclareol, and hydroxycinnamic acids.

In conclusion, the preliminary chemical information obtained, combined with the exerted biological activities, made the investigated neglected matrices a valuable source of functional ingredients, providing a basis for their utilization in various fields.

## Figures and Tables

**Figure 1 plants-12-00795-f001:**
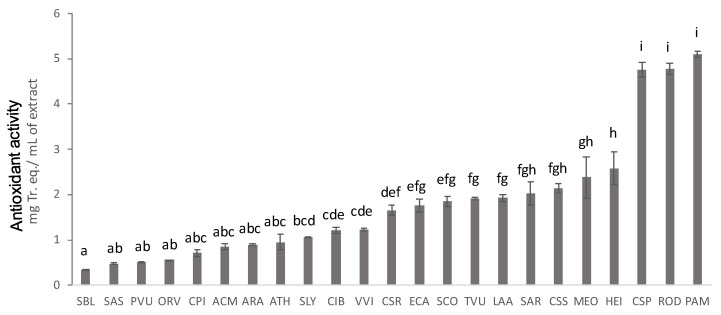
Antioxidant activity expressed in mg of Trolox equivalent/mL of extract, measured by DPPH assay. The samples are indicated on the ‘x’ axis. Different letters within the same assay indicate significant differences in ANOVA tests (*p* < 0.05). Results are expressed as means ± SD of three independent experiments. Samples not reported in this figure showed no detectable activity.

**Figure 2 plants-12-00795-f002:**
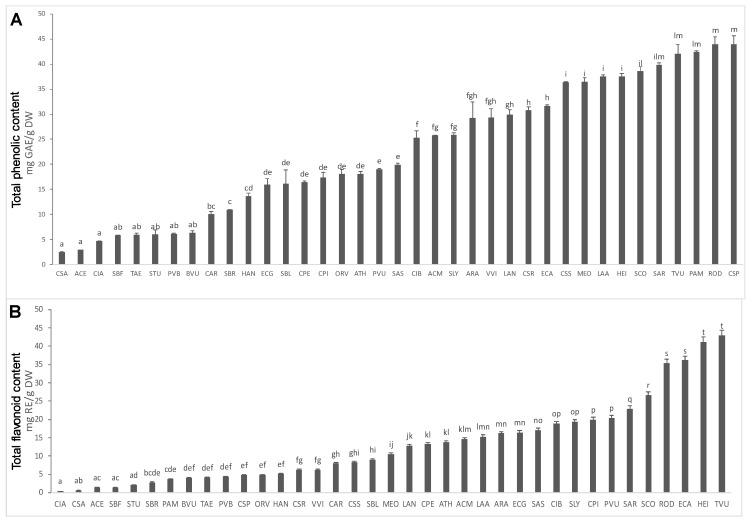
(**A**) Total phenolic content; (**B**) Total flavonoid content. The samples are indicated on the ‘x’ axis, while different letters within the same assay indicate significant differences in ANOVA tests (*p* < 0.05). Results are expressed as means ± SD of three independent experiments.

**Figure 3 plants-12-00795-f003:**
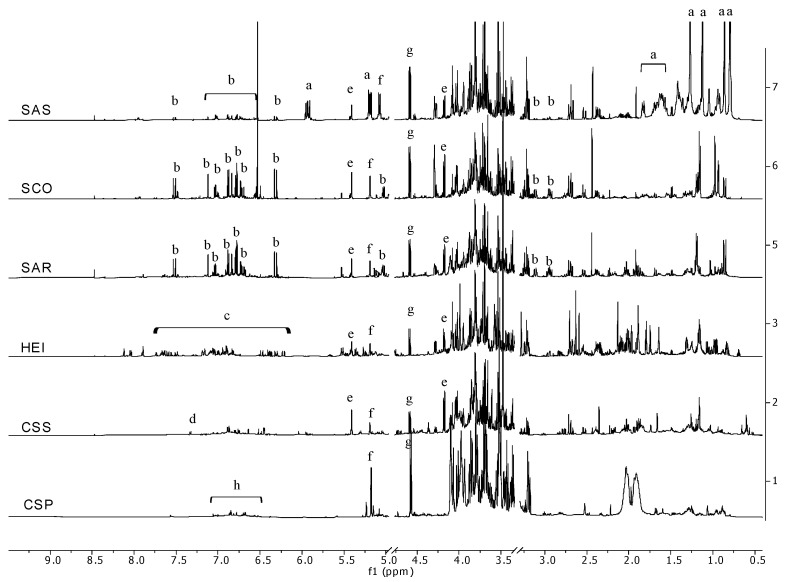
^1^H NMR fingerprinting of the active extracts. a = sclareol, b = rosmarinic acid, c = hydroxycinnamic acids, d = shikimic acid, e = sucrose, f = α-glucose, g = β-glucose, h = tannins.

**Table 1 plants-12-00795-t001:** Plant-neglected matrices investigated in this work, their source (plant scientific and common name and plant part), and used tag are reported together with the type of by-product. Plant scientific names have been updated following the World Checklist of Vascular Plants (WCVP 2020: World Checklist of Vascular Plants). Available online: https://wcvp.science.kew.org/ (accessed on 19 May 2021).

Scientific Name	Common Name	Plant Part	Type of Waste/By-Product	Sample Tag	Voucher Number
*Abutilon theophrasti* Medik.	Velvetleaf	Aerial parts	Pest plant	Ath	WST25
*Achillea millefolium* L.	Common yarrow	Aerial parts	Solid waste from distillation	Acm	WST35
*Allium cepa* L.	Onion	Dry aerial parts	Agricultural residue	Ace	WST1
*Artemisia absinthium* L.	Wormwood	Aerial parts	Solid waste from distillation	Ara	WST36
*Beta vulgaris* L.	Sugar beet	Aerial parts	Agricultural residue	Bvu	WST10
*Camelina sativa* (L.) Crantz	Camelina	Dry aerial parts	Agricultural residue	Csa	WST2
*Castanea sativa* Mill.	Chestnut	Pericarp	Food Industry by-product	Csp	WST16
		Spiny burs	Agricultural residue	Csr	WST3
*Cicer arietinum* L.	Chickpea	Aerial parts	Agricultural residue	Car	WST11
*Cichorium intybus* L.	Chicory	Apical flowering aerial parts	Pest plant	Cia	WST22
		Basal flowering aerial parts	Pest plant	Cib	WST23
*Cucurbita pepo* L.	Courgette	Leaves	Agricultural residue	Cpe	WST12
		Aerial parts	Agricultural residue	Cpi	WST15
*Cupressus sempervirens* L.	Cupressus	Leaves	Solid waste from distillation	Css	WST26
*Echinochloa crus-galli* (L.) P.Beauv.	Cockspur	Flowering aerial parts	Pest plant	Ecg	WST24
*Erigeron canadensis* L.	Horseweed	Aerial parts	Pest plant	Eca	WST21
*Helianthus annuus* L.	Sunflower	Leaves	Agricultural residue	Han	WST4
*Helichrysum italicum* (Roth) G. Don	Curry plant	Aerial parts	Solid waste from distillation	Hei	WST27
*Laurus nobilis* L.	Laurel	Leaves	Solid waste from distillation	Lan	WST28
*Lavandula angustifolia* Mill.	Lavender	Aerial parts	Solid waste from distillation	Laa	WST37
*Melissa officinalis* L.	Lemon balm	Aerial parts	Solid waste from distillation	Meo	WST29
*Origanum vulgare* L.	Oregano	Stems	Food Industry by-product	Orv	WST19
*Phaseolus vulgaris* L.	Bean	Husks	Food Industry by-product	Pvb	WST18
		Aerial parts	Agricultural residue	Pvu	WST13
*Prunus amygdalus* Batsch	Almond	Exocarp and mesocarp	Food Industry by-product	Pam	WST17
*Rosa damascena*	Damask rose	Buds	Solid waste from distillation	Rod	WST30
*Salvia officinalis* L.	Sage	Aerial parts	Solid waste from distillation	Sco	WST33
*Salvia rosmarinus* Schleid.	Rosemary	Aerial parts	Solid waste from distillation	Sar	WST31
*Salvia sclarea* L.	Clary sage	Aerial parts	Solid waste from distillation	Sas	WST32
*Solanum lycopersicum* L.	Tomato	Basal leaves	Agricultural residue	Sly	WST14
*Solanum tuberosum* L.	Potato	Leaves	Agricultural residue	Stu	WST8
*Sorghum bicolor* (L.) Moench	Sorghum	Leaves	Agricultural residue	Sbl	WST6
		Roots	Agricultural residue	Sbr	WST7
		Stems	Agricultural residue	Sbf	WST5
*Thymus vulgaris* L.	Common thyme	Aerial parts	Solid waste from distillation	Tvu	WST34
*Triticum aestivum* L.	Wheat	Dry aerial parts	Agricultural residue	Tae	WST9
*Vitis vinifera* L.	Grape	Pomace	Food Industry by-product	Vvi	WST20

## Data Availability

Not applicable.

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
