# Peer review of "Phytochemical Profile and In Vitro Bioactivities of Plant-Based By-Products in View of a Potential Reuse and Valorization"

_plants, 2023, doi:10.3390/plants12040795_

Round 1

Reviewer 1 Report

Lines 47-50; should be moved to the end of the introduction.

lines 59-60; In addition------determined. should be moved to the aim of the study.

lines 73-75; and lines 81-82 these are the objectives of the study with the above sentences, i suggest reshaping and writing at the end of the introduction.

Table S1; the statistical differences are not shown.

Figures 1, 2; both X and Y should be labelled.

line 106; A typographical error.

line 109; the unit of the test was written as 100 µl/mL while in table S1 was written as µg/mL, so, please use the same unit throughout the manuscript.

Line 147, Figure 3 can you quantify the active compounds?

Line 266; how did you perform statistical analysis of the data while you repeated experiment two times?

Table S1 should be shown the name of the parameters rather than just the unit.

Reviewer 2 Report

The work describes in vitro antioxidant and anti-tyrosinase activity together with the antibacterial activity of some neglected plant matrices against Clavibacter michiganensis. The paper is well written and organized. It falls fully within the scope of the Journal and in particular within the Special iusse “New Trends in Plant Science in Italy”. According to me the paper can be published after minor remarks:

1) In Figure 1: title of the ordinate axis should be “mg Tr. eq/ mL of extracts” not “ TE/mL…”

2) In References: in ref. n. 14 “Salvia miltiorrhiza” should be in italics form (line 324)

3) In References: in ref. n. 15 “sardinia” should be “Sardinia” (line 327)

4) In lines 21,23, 109, 112, 113, 126 and 241 “ml” should be “mL”.

5) 3.8 NMR and ESI-MS analysis: The mass spectrometry method is described in M&M but then not mentioned in the Results and Discussion. Perhaps these analysis were only served to verify the correct characterization of the identified metabolites by NMR. If so, add an explanation in the text, otherwise you can remove the MS technique from the paper.

Reviewer 3 Report

Dear Author,

Firstly, Phytochemical profile, as mentioned in the manuscript tile but no analysis regarding the phytochemical composition of the studied extracts neither the most actives one was done.

ESI-MS: While mention in materials and methods section, no single word related to ESI-MS analysis in result section, which is not acceptable to give detail for any experiment, but no result given in the manuscript neither in supplementary data.

In Plant material and sample treatment part you used many different Plant part (Aerial parts, Dry aerial parts, Pericarp, Spiny burs, Apical flowering, Basal flowering, Leaves, Stems, Husks, Exocarp, Buds, Basal leaves, Roots, Pomace) obtained from different  type of waste/by-products (Pest plant, Solid waste from distillation, Agricultural residue, Food Industry by-products, ) but you used the same extraction protocol: I think it would be better to use different protocols for each type of waste/by-product and also in each plant parts. I think each samples need specific protocol to extract the bioactive compounds.

Although the number of tested samples is high, the conducted experiments is very few and limited also it need specific protocols for each sample in order to extract the potent bioactive molecules. In fact, antioxidant activity (just DPPH essay) and total polyphenol and flavonoid contents is not enough to evaluate the phytochemical of the studied extracts.

For the antibacterial activities, it would be better to present the results in the form of a table, to help readers to follow the results of the different active extracts (MIC, controls, statistic results,….).

In the Results and Discussion section (About 47 lines): only 2 references were used to discuss the obtained results, which makes a very poor discussion. I strongly suggest to add an explanation about the obtained results and compare with the literature.  

I suggest conducting LC-MS or at least GC-MS to identify the presence of potent compounds in the active extracts.

Round 2

Reviewer 1 Report

It is now can be accepted.

Reviewer 3 Report

Dear author,

Thank you for your effort and your answers